# The Effects and Regulatory Mechanism of Casein-Derived Peptide VLPVPQK in Alleviating Insulin Resistance of HepG2 Cells

**DOI:** 10.3390/foods12132627

**Published:** 2023-07-07

**Authors:** Dapeng Li, Jianxin Cao, Jin Zhang, Tong Mu, Rubin Wang, Huanhuan Li, Honggang Tang, Lihong Chen, Xiuyu Lin, Xinyan Peng, Ke Zhao

**Affiliations:** 1College of Life Science, Yantai University, Yantai 264005, China; ldpyantaidaxue@163.com; 2Zhejiang Academy of Agricultural Sciences, Hangzhou 310021, China; m17809297329@163.com (J.C.); zhangjin@zaas.ac.cn (J.Z.); mutong025@163.com (T.M.); m18267588225@163.com (R.W.); huanhuanlee325@163.com (H.L.); zaastang@163.com (H.T.); cwc528@163.com (L.C.); 13127410550@163.com (X.L.); kzhao@snnu.edu.cn (K.Z.); 3College of Food Engineering and Nutritional Science, Shaanxi Normal University, Xi’an 710062, China

**Keywords:** VLPVPQK, antioxidant activity, HepG2, glucose uptake, transcriptome

## Abstract

The liver plays a key role in keeping the homeostasis of glucose and lipid metabolism. Insulin resistance of the liver induced by extra glucose and lipid ingestion contributes greatly to chronic metabolic disease, which is greatly threatening to human health. The small peptide, VLPVPQK, originating from casein hydrolysates of milk, shows various health-promoting functions. However, the effects of VLPVPQK on metabolic disorders of the liver are still not fully understood. Therefore, in the present study, the effects and regulatory mechanism of VLPVPQK on insulin-resistant HepG2 cells was further investigated. The results showed that VLPVPQK exerted strong scavenging capacities against various free radicals, including oxygen radicals, hydroxyl radicals, and cellular reactive oxygen species. In addition, supplementation of VLPVPQK (62.5, 125, and 250 μM) significantly reversed the high glucose and fat (30 mM glucose and 0.2 mM palmitic acid) induced decrement of glucose uptake in HepG2 cells without affecting cell viability. Furthermore, VLPVPQK intervention affected the transcriptomic profiling of the cells. The differentially expressed (DE) genes (FDR < 0.05, and absolute fold change (FC) > 1.5) between VLPVPQK and the model group were mostly enriched in the carbohydrate metabolism-related KEGG pathways. Interestingly, the expression of two core genes (HKDC1 and G6PC1) involved in the above pathways was dramatically elevated after VLPVPQK intervention, which played a key role in regulating glucose metabolism. Furthermore, supplementation of VLPVPQK reversed the high glucose and fat-induced depression of AKR1B10. Overall, VLPVPQK could alleviate the metabolic disorder of hepatocytes by elevating the glucose uptake and eliminating the ROS, while the HKDC1 and AKR1B10 genes might be the potential target genes and play important roles in the process.

## 1. Introduction

Oxidative damage led to a variety of chronic diseases such as cancer, diabetes, inflammation, cardiovascular disease, asthma, and Alzheimer’s disease [1,2]. Diabetes was characterized by chronic hyperglycemia, while high levels of reactive oxygen species (ROS) were generated during the glycolytic reaction under a hyperglycemic state. Excessive ROS could oxidatively damage pancreatic β-cells and cause endothelial dysfunction [3], which then exacerbates the disorder of glucose metabolism and brings a great threat to human health [4]. Considering the side effects of therapeutics for patients with those chronic diseases, the exploration of bioactive substances from food resources for the prevention of related metabolic disorders attracted more and more researchers’ interest.

Bioactive peptides originated from food protein exerted multiple health-promoting functions, and were considered to be safe and efficient [5,6]. Milk is a major source of nutrients for humans, which owns a high quantity and quality of protein and is an important resource for the generation of bioactive peptides. It is reported that milk-derived bioactive peptides participated in the regulation of human physiology, nervous development, gastrointestinal, cardiovascular, and immune systems [7,8,9]. Nowadays, bioactive peptides were normally clustered into different groups based on their biological functions, such as DPP-IV inhibitory peptides, ACE inhibitory peptides, antioxidant peptides, antimicrobial peptides, etc. Among them, the antioxidant peptides attracted the interest of researchers in food and pharmaceutical fields, some of them exert a wide range of biological properties, including antioxidant, anti-hypertension, immunomodulation, and alleviation of insulin resistance, high cholesterol/glucose, etc. [10,11,12,13].

The VLPVPQK was a β-casein-derived antioxidant peptide, which possessed potent antioxidant activity [14,15,16]. Savita Devi et al. [17] investigated the antioxidant properties of VLPVPQK by deleting its C-terminal sequence, and the results demonstrated that glutamine and lysine were essential amino acids accounting for the antioxidant, anti-inflammatory, and cytoprotective effects of VLPVPQK. In addition, it has been reported that VLPVPQK could attenuate apoptosis, dysfunction, and oxidative damage in osteoblast and fibroblast cells [18,19,20]. Furthermore, VLPVPQK showed good anti-osteoporotic effects in rats by binding with calcium, elevating bone density/strength, and reducing the expression of bone resorption-related cytokines [21,22]. Interestingly, the intact VLPVPQK could be partially transported across the cell membrane [23]. The liver plays a key role in glucose and lipid metabolism, and the dysfunction of the liver contributes greatly to chronic metabolic syndromes, such as diabetes and insulin resistance. The oxidative status and excessive ROS could induce morphological changes and functional disorders in mammalian cells [24]. Considering the potential antioxidant activity reported previously, it could be deduced that VLPVPQK might exert protective effects on the liver, thereby alleviating the related metabolic disorders.

Therefore, in the current study, the antioxidant capacity and possible regulatory effects of VLPVPQK on the high fat and sugar-induced dysfunction of HepG2 cells were first evaluated. Then, the RNAseq method was applied to further explore its possible regulatory mechanism. The information obtained from this study gave insight into the biological function of VLPVPQK, and provided the basis for its application in functional food or pharmaceutical materials.

## 2. Materials and Methods

### 2.1. Materials

Goat milk-derived antioxidant peptide VLPVPQK with 98% purity was synthesized by Shanghai QiangYao Biology. Human hepatoma cells (HepG2) were purchased from the stem cell bank of the Chinese Academy of Sciences. MEM cultures and fetal bovine serum were supplied by GIBCO Invitrogen Co. All the other reagents used in this study were the highest purity commercially.

### 2.2. Antioxidant Activity Measurement

#### 2.2.1. Oxygen Radical Absorbance Capacity (ORAC)

The ORAC was determined according to the method of Huang et al. with some modifications [25]. Briefly, AAPH solution (40 mM) was added to different concentrations of VLPVPQK solution and OD values were measured every 3 min for a total of 120 min at an excitation wavelength of 485 nm and an emission wavelength of 535 nm. Trolox was used as a positive control. Three parallel experiments were carried out for each concentration. The formula was shown as follows:AUC = 0.5 × [2 × (*A*_0_ + *A*_1_ +…+ *A*_n−1_ + *A*_n_) − *A*_0_ − *A*_n_] × Δt(1)
where *A*_n_ is the fluorescence value at time n.

Net AUC = AUC_Sample_ − AUC_blank_ was used to calculate the oxidative radical uptake capacity of the sample and the standard (Trolox).

#### 2.2.2. Hydroxyl Radical Scavenging Capacity

Following the method reported previously [26], 25 μL of VLPVPQK solution (0.2, 1, and 5 mM) was added to an equal volume of the mixed solution including 1.04 mM disodium EDTA, 10 mM hydrogen peroxide, 2 mM ascorbic acid, 60 mM deoxyribose, and 1 mM ferric chloride. The reaction was terminated by adding 250 μL of TCA (2.8% by mass) after incubation at 37 °C for 1 h. Then, 250 μL TBA (1% mass fraction) was added to the test tube and boiled for 15 min, and the absorbance value was measured at 532 nm at room temperature. The formula was calculated as follows:scavenging rate% = (1 − *A*_Sample_/*A*_blank_) × 100(2)

### 2.3. Cellular Experiments

#### 2.3.1. Cell Viability

HepG2 cells were incubated with MEM medium containing 10% fetal bovine serum (FBS), 100 units/mL penicillin/streptomycin in an incubator with 5% CO_2_ at 37 °C. The cells were seeded into 96 well culture plates at the concentration of 1 × 10^5^ cells/mL, and incubated for 48 h. Then the cells were treated with different concentrations of VLPVPQK (0, 3.91, 7.81, 15.63, 62.5, 125, 250, and 500 μM), or 0.2 mM metformin (Met), while the Met group was used as a positive control. After 48-h treatments, 0.5 mg/mL thiazole blue was added and incubated at 37 °C for another 4 h, then 150 μL of DMSO was used to dissolve the precipitate. The optical density was measured at 570 nm. Three sets of parallel experiments were set up for each experiment. The cell viability was calculated using the following formula:Cell viability (%) = OD_sample 570_/OD_control 570_ × 100%(3)

#### 2.3.2. Cell Culture and Treatment

After reaching 80% confluence, the cells were starved in a MEM medium containing 2% FBS for 24 h. Then, except for the blank group, cells in all the other groups were treated with 30 mM glucose plus 0.2 mM palmitic acid (PA) for 48 h. Then the control, model, Met, and VLPVPQK groups were treated with growth media, 30 mM glucose plus 0.2 mM palmitic acid (PA), 2 mM metformin (Met) dissolving in growth media [27], and different concentrations (62.5, 125, 250 μM) of VLPVPQK peptides dissolving in growth media for another 48 h, respectively.

#### 2.3.3. Glucose Uptake and ROS Assessment

After 48-h treatments described above, the culture supernatant was removed. Then 10 μM 2-NBDG or DCFH-DA (dissolved in fresh culture medium) were added into the cells and incubated at 37 °C for another 30 min. The 2-NBDG is a fluorescently labelled analogue of 2-deoxyglucose and is commonly used as a tracer for cellular glucose metabolism [28]. After the supernatant was abandoned, the cells were washed 3 times with phosphate-buffered saline (PBS) and harvested by centrifugation. The 2-NBDG fluorescence was detected by flow cytometry (cytoFLEXS, Beckeman Coulter, Pasadena, CA, USA). The image of the cells treated with DCFH-DA was captured using a fluorescent microscope (excitation wavelength 488 nm, emission wavelength 525 nm), and the density of fluorescence was calculated by Image J software.

#### 2.3.4. RNA-Seq and Data Analysis

The possible regulatory mechanism of VLPVPQK was investigated using the RNA-seq method. Total RNA was extracted with Trizol reagent according to the manufacturer’s instructions, and genomic DNA was removed using DNaseI (TaKara). The quality and integrity of the RNA was detected by NanoDrop ND-2000 (Thermo Scientific, Waltham, MA, USA) and 2100 Bioanalyzer (Agilent Technologies, Palo Alto, CA, USA). RNA libraries were prepared by application of TruSeq™ RNA Sample Preparation Kits from Illumina (Carlsbad, CA, USA), then the cDNA was synthesized using the SuperScript double-stranded cDNA synthesis kit (Invitrogen, Shanghai, China). After enrichment by PCR and quantification by TBS380 (picogreen), the high-throughput sequencing was performed using the IlluminaHiSeq X10 (2 × 150 bp read length). After a quality screening of the raw sequencing data, the expression levels of genes and transcripts were quantified using the software RSEM. The differentially expressed (DE) genes were analyzed by DESeq2 with the absolute fold change (FC) > 1.5 and false discovery rate (FDR) < 0.05. Then the Gene Ontology database (GO) (http://www.geneontology.org/, accessed on 7 December 2021) and the Kyoto Encyclopedia of Genes and Genomes (KEGG) (http://www.genome.jp/kegg/, accessed on 7 December 2021) were used for functional enrichment of the DE genes in free online platform of Majorbio Cloud Platform (www.majorbio.com, accessed on 7 December 2021), while the *p* < 0.1 corrected by the Benjamini-Hochberg test was considered as significantly enriched function. while the *p* < 0.1 corrected by the Benjamini-Hochberg test was considered as significantly enriched function.

### 2.4. Statistical Analysis

GraphPad Prism (version 8.0) was applied for the data analysis, and all the results were expressed as means ± standard of error mean (SEM). The data were analyzed using one-way analysis of variance (ANOVA) followed by Dunnetts multiple comparison post-test. Statistical significance was set at *p* < 0.05. All the measurements were performed in triplicate.

## 3. Results

### 3.1. Assay of Antioxidant Activity

As shown in Figure 1A, the ORAC net fluorescence rates were elevated from 27.56 ± 6.9% to 115.44 ± 7.84% with the increasing concentrations of VLPVPQK from 2 to 10 mM, in a dose-dependent manner. Similarly, the hydroxyl radical scavenging rates of the cells were increased after treatment with VLPVPQK (0.2, 1, and 5 mM), with the lowest value (4.4 ± 1.51%) at 0.2 mM and the highest value (29.9 ± 0.62%) at 5 mM (Figure 1B).

### 3.2. Cell Viability of HepG2 Cells

As shown in Figure 2A, compared with the control group, supplementation of VLPVPQK at concentrations lower than 250 μM did not affect the cell viability (*p* > 0.05), while 500 μM VLPVPQK significantly decreased the cell viability (*p* < 0.05). Similar results were observed when the VLPVPQK groups were compared to the Met group.

### 3.3. Effect of VLPVPQK on Intracellular Glucose Consumption and ROS Level

As shown in Figure 2B, compared with the blank group, the ROS content in the cells was significantly increased in the model group (*p* < 0.05). In addition, the metformin intervention reversed the high glucose plus fat-induced increment of ROS (*p* < 0.01). Compared with the model group, the ROS contents decreased, dose-dependently, with the concentrations of VLPVPQK increasing from 62.5 to 250 mM, while the 62.5 mM and 250 mM VLPVPQK showed significant differences (*p* < 0.01).

Compared with the blank group, high glucose plus fat treatment dramatically decreased glucose uptake (*p* < 0.01). Supplementation of metformine (2 mM) or VLPVPQK (62.5, 125, and 250 mM) significantly elevated the amount of glucose uptake compared with the model group (*p* < 0.01) (Figure 2B,C). In addition, the amount of glucose uptake in the cells treated with different concentrations of VLPVPQK (62.5, 125, and 250 mM) was higher than that treated with metformine (*p* < 0.05).

### 3.4. VLPVPQK-Regulated Transcriptomic Profiling of HepG2 Cells

A total of 512,264,394 raw reads were detected in the 9 cell samples, and an average of 56.89 ± 1.77, 56.57 ± 3.13, and 57.29 ± 1.96 million reads were detected in blank, model, and VLPVPQK groups, respectively. After mapping to the human genome, 13,666, 13,790, and 13,829 genes were detected in blank, model, and VLPVPQK groups, respectively (Figure 3A). Among those genes, most of them (*n* = 12,423) were commonly expressed in the three groups. The results of principal component analysis (PCA) revealed that the transcriptomic profiling in the blank, VLPVPQK, and model groups had clear clustering (Figure 3B). As shown in Appendix A, 9012 DE genes (absolute FC > 1.5, and FDR < 0.05) were observed between the blank and model groups (including 4570 up-regulated and 4442 down-regulated genes), and 69 DE genes (absolute FC > 1.5, and FDR < 0.05) were found between the model and VLPVPQK groups (including 47 up-regulated and 22 down-regulated genes).

### 3.5. Functional Enrichment of DE Genes

The GO (Gene Ontology) and KEGG (Kyoto Encyclopedia of Genes and Genomes) pathway enrichment was performed to clarify the function of those DE genes. The top 5 GO terms of biological processes enriched by DE genes between the blank and model groups were “nucleotide-excision repair, DNA incision”, “intraciliary transport involved in cilium assembly”, “cellular component maintenance”, “negative regulation of coagulation”, and “transcription initiation from RNA polymerase I promoter” (Figure 3C) (*p* < 0.05). On the other hand, the most enriched biological processes enriched by DE genes between the model and VLPVPQK groups were “homeostatic process”, “regulation of biological quality”, “organic cyclic compound metabolic process”, “metabolic process”, and “primary metabolic process” (Figure 3D) (*p* < 0.05).

As shown in Figure 4A, among the top 20 KEGG pathways that were enriched by the DE genes between the blank and model groups (*p* adjust value < 0.05), the “Metabolic pathways” were the most enriched pathway with 666 genes involved in the process. On the other hand, there were 11 KEGG pathways enriched using the DE gene between the model and VLPVPQK groups (Figure 4B) (*p* adjust value < 0.05). Interestingly, 8 out of the 11 pathways were associated with carbohydrate metabolism, including “Galactose metabolism”, “Starch and sucrose metabolism”, “Amino sugar and nucleotide sugar metabolism”, “Fructose and mannose metabolism”, “Carbohydrate digestion and absorption”, “Glycolysis/Gluconeogenesis”, “Insulin resistance” and “Insulin signaling pathway”. In addition, similar to the blank and model groups, the majority of DE genes between the model and VLPVPQK groups took part in the “Metabolic pathways” (Figure 4B).

### 3.6. KEGG Pathways and Genes Related to Carbohydrate and Metabolic

To investigate the possible effects and regulatory mechanisms of the antioxidant peptide VLPVPQK on the high fat and glucose-induced disorder of liver cells. The 9 KEGG pathways related to carbohydrate metabolism were selected for further analysis, and the expression profile and functional interaction of genes involved in those pathways were investigated. There were 7 genes (HKDC1, TRIB3, GNE, AKR1B10, CA9, G6PC1, and NDUFA4L2) involved in those pathways (Figure 4C). As shown, the expressions of HKDC1, TRIB3, GNE, AKR1B10, and CA9 were depressed in the model group and elevated after VLPVPPQK intervention, while those of NDUFA4L2 showed a reverse pattern. In addition, the gene expressions of G6PC1 did not show significant differences between the blank and model groups, while supplementation of VLPVPPQK dramatically stimulated the expressions of G6PC1 compared with the model group. Furthermore, the expressions of HKDC1, AKR1B10, and TRIB3 showed relatively high abundance (average TPM > 10). As shown in Figure 4D, HKDC1 and G6PC1 had 7 edges with 9 carbohydrate metabolism and insulin signaling-related pathways. AKR1B10 was involved in “Fructose and mannose metabolism”, “Galactose metabolism” and “Metabolic pathways”. Genes associated with “metabolic pathways” also included GNE, CA9, and NDUFA4L2, while TRIB3 was involved in “insulin resistance”.

## 4. Discussion

The liver, one of the key target tissues affected by insulin, plays an important role in maintaining the metabolic homeostasis of glucose and lipid [29]. Long-term high nutrient intake led to glucolipotoxicity through inflammatory factors, reactive oxygen species, and free radicals, and contributed greatly to the metabolic disorders of the liver, such as fatty liver or diabetes [30,31]. The goat β-casein-derived antioxidant peptide VLPVPQK showed strong antioxidant capacity reflected by the results of ORAC and hydroxyl radical scavenging capacity. Considering the toxicity of free radicals, it is reasonable to deduce that the peptide VLPVPQK might exert an important role in alleviating the metabolic disorder of the liver.

It was reported that intracellular ROS levels rose significantly in the presence of insulin resistance (IR) [32,33]. The hyperglycemia-induced pentose phosphate pathway impairment, NADPH reduction, and NADPH oxidase activation were closely related to glucose metabolism and ROS generation [34,35]. The ROS, as cell signaling molecules, exert vigorous oxidative activity; however, the excessive ROS can disrupt the antioxidant defence of tissues and finally lead to oxidative stress [36,37]. Under oxidative stress conditions, ROS can damage cellular proteins, lipids, and DNA, leading to cellular damage that further results in a variety of pathologies, such as ageing, cancer, cardiovascular disease, and diabetes [38]. HepG2 cells had been widely used to investigate the pathogenesis of IR and the hypoglycemic effects of bioactive substances in vitro [13,39,40]. We successfully established an IR cell model, which was exposed to high glucose (30 mM) and high fat (0.2 mM palmitate) as described previously [41]. In the current study, VLPVPQK intervention decreased high glucose plus the fat-induced elevation of ROS content, thereby attenuating oxidative stress and contributing to the alleviation of insulin resistance. The biological activities of peptides are closely related to their hydrophobicity, charge, and biological interactions [42,43], while the hydrophobic amino acids (leucine, glutamine, and lysine) and the positive charge of VLPVPQK might account for its antioxidative characteristics. In addition, it has been reported that the C-terminal glutamine and lysine played key roles in the radical scavenging and injury-protecting activities of VLPVPQK in rat fibroblast cells [17]. However, the relationship between the structure and biological function of VLPVPQK in liver cells still requires further research.

Except for the free radicals, the disorder of glucose metabolism in the liver contributed greatly to insulin resistance. Under normal circumstances, insulin stimulates glucose uptake and glycogen synthesis in hepatocytes, while insulin resistance affects glucose storage (glycogenesis) and glucose output (gluconeogenesis), thereby causing lipid metabolic disorders and type 2 diabetes [44,45]. Therefore, improving the insulin resistance in the liver is an effective way to prevent and treat type 2 diabetes. Hepatic insulin resistance is characterized by depressed glucose consumption and storage, which are accounted partially for the elevated circulating glucose level and disorder of lipid synthesis [46,47]. In the present study, the VLPVPQK enhanced glucose uptake in insulin-resistant HepG2 cells in vitro, which demonstrated that VLPVPQK could reverse the insulin resistance-induced depression of glucose consumption. Interestingly, the glucose uptake in the cells treated with VLPVPQK was higher than that treated with metformin, which was an oral medicine widely used to treat type 2 diabetes. The above results demonstrated the potential hypoglycemic activity of VLPVPQK. However, the concentration-dependent manner regarding glucose uptake was not observed in the range of 62.5 to 250 μM of VLPVPQK. The dose effect was also not observed in another bioactive substance Loureirin B, which attenuated IR in HepG2 cells by elevating glucose uptake and consumption [33]. Because lots of pathways, such as IRS1/AKT/GLUT4, PI3K/Akt, and AMPK signaling pathways were related to glucose metabolism [48,49,50,51], the complex regulatory mechanisms might explain the dose-independent phenomena of glucose uptake. While the effects of VLPVPQK on those pathways attracted our attention in the follow-up studies.

To explore the possible mechanism of VLPVPQK in regulating the IR-induced metabolic disorder of the liver, the genome-wide gene expressions of HepG2 cells that were modified by high glucose plus fat and/or VLPVPQK were further investigated. It was found that the high glucose plus fat and VLPVPQK intervention affected the transcriptomic profiling of HepG2 cells. Most majority of the DE genes in model vs. blank groups or VLPVPQK vs. model groups were enriched to metabolic pathway, indicating metabolism might be the major regulatory target of IR in the liver. Interestingly, the top six KEGG pathways enriched by DE genes between VLPVPQK and model groups were involved in carbohydrate metabolism. Glucose is the major source of energy and macro-molecular building blocks, together with the above mentioned elevation of glucose uptake after VLPVPQK intervention, it is reasonable to deduce that VLPVPQK might participate in the alleviation of IR by regulating glucose metabolism in the liver. Therefore, the carbohydrate metabolism and insulin signaling-related pathways were further analyzed, which might give useful clues about the regulatory mechanism of VLPVPQK at the gene expression level.

Among the 7 genes involved in carbohydrate metabolism and insulin signaling-related pathways, the hexokinase domain containing 1 (HKDC1) and glucose-6-phosphatase catalytic subunit 1 (G6PC1) genes were considered as core genes, as they participated in 7 out of the 9 pathways mentioned above. Hexokinase catalyzes hexose phosphorylation and regulates the rate-limiting step of glucose metabolism [52], while the HKDC1 had recently been discovered as the fifth hexokinase, which contributed to whole-body glucose disposal, insulin sensitivity, and nutrient balance [53,54,55]. It was reported that HKDC1 was expressed in the liver, and metabolic stress could reduce the HKDC1 expression [55,56]. Consistently, HKDC1 was the most abundantly expressed hexokinase in HepG2 cells (Appendix A), and the high glucose plus fat induced-IR depressed the expression of HKDC1 in this study. HKDC1^+/−^ mice exhibited impaired glucose tolerance and decreased liver triglycerides/glycogen levels [57], while over-expression of HKDC1 could improve systemic glucose tolerance and insulin sensitivity in pregnant mice [55]. Those studies suggested that HKDC1 was important for glucose utilization and hepatic energy storage. In the current study, the VLPVPQK intervention reversed the high glucose plus fat-induced restraint of glucose uptake and HKDC1 expression. Therefore, it is reasonable to deduce that VLPVPQK might alleviate insulin resistance in HepG2 cells by stimulating glucose uptake, and the HKDC1 gene might play an important role in the process. Considering the inconsistency among gene expression, protein expression, and enzyme activity, the specific relationship between VLPVPQK and HKDC1 regarding glucose metabolism requires further research.

On the other hand, the G6PC1 gene encodes the catalytic unit of glucose-6-phosphatase (G6Pase), which catalyzes the glucose-6-phosphate produced from glycogen or gluconeogenic precursors to glucose and is the key enzyme in the homeostatic regulation of blood glucose levels [57,58]. Three isoforms of G6PC_S_ (G6PC1-3) had been identified, while G6PC1 was mainly expressed in the liver and kidney and acted as the gatekeeper of glucose production [57]. In the current study, VLPVPQK supplementation stimulated the expression of G6PC1 compared to the model group. Elevated G6PC1 mRNA was observed in animal models of diabetes, and the over-expression of G6PC1 elevated hepatic glucose production [59,60,61]. Therefore, the elevated expression of G6PC1 induced by VLPVPQK might lead to the increasing generation of glucose in the liver. However, the actual rate of glucose production was determined by the coupling of G6PC1 with SLC37A4 [62,63]. The SLC37A4 is a G6P transporter, which is responsible for the access of G6P to the G6PC1 active site in the endoplasmic reticulum [64]. The transport of G6P from cytosol was considered as the rate-limiting step for glucose production [61,63]. In the current study, the expression of SLC37A4 was not affected before and after VLPVPQK treatment (Appendix A). Overall, it was suggested that VLPVPQK intervention might not take part in the regulation of glucose production in liver cells.

In addition, the aldo-keto reductase family 1 member B10 (AKR1B10) gene attracted our attention, which showed relatively high abundance in HepG2 cells (with an average TPM of 51.2 ± 7.5) and was involved in 3 KEGG pathways including “Glucose metabolism”, “Fructose and mannose metabolism”, and “Metabolic pathways”. The AKR1B10 played important roles in protecting host cells against dietary and lipid-derived cytotoxic carbonyls [64,65]. It was found in this study that the antioxidant peptide VLPVPQK could reverse the high glucose and fat-induced depression of AKR1B10. Consistently, another antioxidant, ethoxyquin, could up-regulate the expression of AKR1B10 [66]. Furthermore, the antioxidant response elements were found in the promoter of the AKR1B10 gene [67]. Overall, it was suggested that the AKR1B10 gene might be regulated by antioxidants, and VLPVPQK might exert protective effects on liver cells by elevating the AKR1B10 gene expression. However, genetic silencing of the AKR1B10 gene or inhibition of AKR1B10 activity contributed to the alleviation of non-alcoholic fatty liver disease [68,69]. Considering the multiple roles of AKR1B10 and different animals/tissues used in the above research, the specific role of AKR1B10 in the insulin-resistant HepG2 cells and its relationship with VLPVPQK require further research.

Overall, VLPVPQK could alleviate the metabolic disorder of hepatocytes by eliminating the ROS and elevating glucose uptake, while the glucose metabolism-related gene HKDC1 and toxic-converting gene AKR1B10 might be the potential target genes, and play important roles in the process. Nevertheless, the limitation existed in this study due to the limited number of genes included and the lack of validation on HKDC1 and AKR1B10 genes. The proteomics and gene interference method will be included in our future study to obtain more related proteins and the specific roles of the target genes.

## Figures and Tables

**Figure 1 foods-12-02627-f001:**
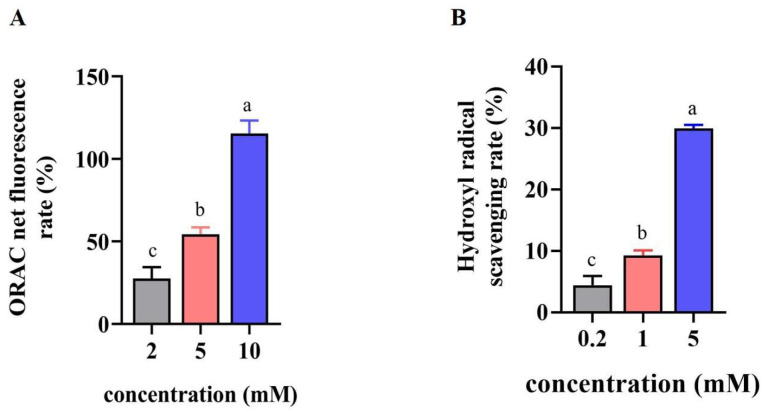
Antioxidant activity of VLPVPQK. (**A**) ORAC net fluorescence rate of HepG2 cells at different concentrations of VLPVPQK. (**B**) Hydroxyl radical scavenging rate of HepG2 cells at different concentrations of VLPVPQK. Data were presented as mean ± SEM (*n* = 3), and statistical analysis was followed by one-way ANOVA. Different letters represent *p* < 0.05.

**Figure 2 foods-12-02627-f002:**
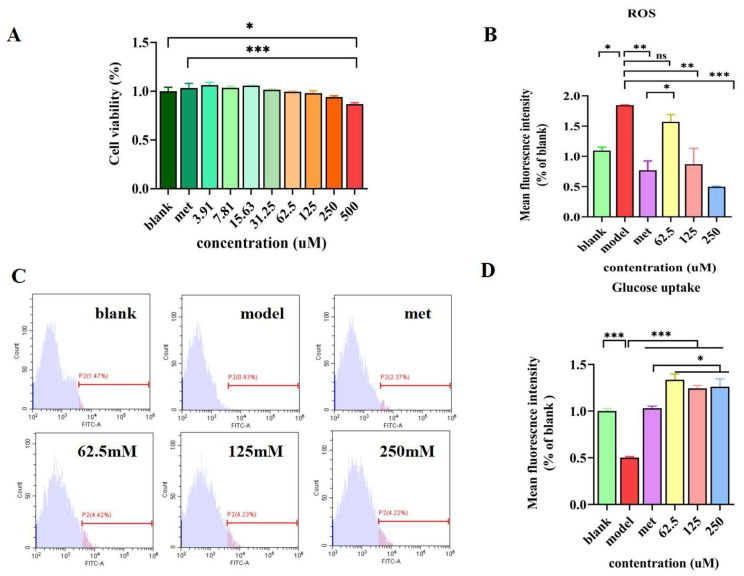
The effects of VLPVPQK on cell viability, ROS content, and glucose uptake. (**A**) Cell viability of HepG2 cells. (**B**) ROS content of HepG2 cells. (**C**) Histogram of glucose uptake of HepG2 cells. (**D**) Glucose uptake was assessed by flow cytometry after cells were incubated with 2-NBDG or DCFH-DA. Data were presented as mean ± SEM (*n* = 3), and statistical analysis was followed by one-way ANOVA. Different letters represent *p* < 0.05. * *p* < 0.05, ** *p* < 0.01, *** *p* < 0.001; ns: not significant.

**Figure 3 foods-12-02627-f003:**
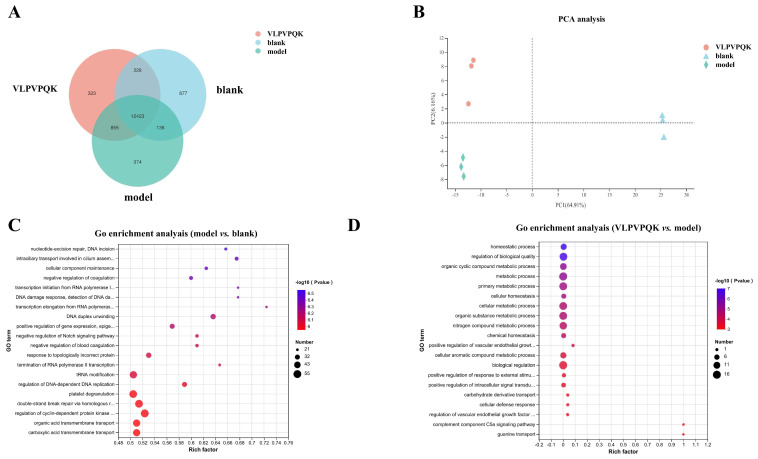
Gene expression profiling and GO term analysis in HepG2 cells among three groups. (**A**) Venn diagrams of expressed genes in the blank, model, and VLPVPQK groups. (**B**) Principal component analysis of expressed genes for all the samples. (**C**) GO enriched terms in biological processes affected by high fat plus glucose treatment. (**D**) GO enriched terms in biological processes affected by VLPVPQK supplementation. (*n* = 3).

**Figure 4 foods-12-02627-f004:**
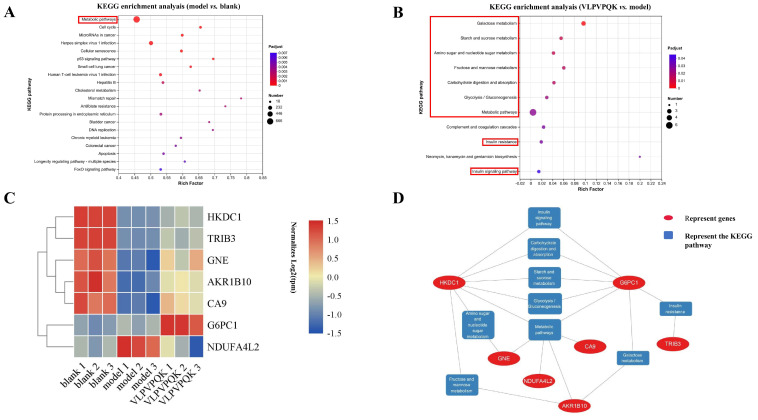
KEGG pathways and differentially expressed (DE) genes analysis in HepG2 cells among three groups. (**A**) KEGG pathways affected by high fat plus glucose treatment. (**B**) KEGG pathways affected by VLPVPQK supplementation. (**C**) Heatmap of DE genes associated with carbohydrate metabolism. (**D**) Interaction network of selected DE genes and significant KEGG pathways related to carbohydrate metabolism. (*n* = 3).

## Data Availability

Data is contained within the article or supplementary material.

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
