# Peer review of "The Effects and Regulatory Mechanism of Casein-Derived Peptide VLPVPQK in Alleviating Insulin Resistance of HepG2 Cells"

_foods, 2023, doi:10.3390/foods12132627_

Round 1

Reviewer 1 Report

It was an interesting read. I think there is a high quality experiment developing.

  1. You are shown in Figure 2A, "compared with the blank group, ROS content was signifi-181 cantly increased after treated with high fat plus glucose in the model group (p < 0.05)."
Please discuss specifically how there may be a relationship between ROS and glucose uptake, but on what grounds this can be determined, citing the results of human clinical trials as well as Hep G2cells.

2. I feel that the data are not sufficient to determine that increased expression of the HKDC1 and G6PC1 genes is associated with increased glucose uptake in HepG2 cells. Not only phenomenology, but also a variety of examples and the increase in the number of proteins involved should be investigated, and it is necessary to fully reconsider whether such a conclusion can be drawn from this experiment alone.

Author Response

Thanks so much for your hard work on our manuscript "foods-2435865". We thank the reviewers for thoroughly reviewing our manuscript and making thoughtful comments. The manuscript has been revised carefully, and the detailed corrections requested by the reviewers have been done and listed below point by point.

  1. You are shown in Figure 2A, "compared with the blank group, ROS content was significantly increased after treated with high fat plus glucose in the model group (p < 0.05)."

Please discuss specifically how there may be a relationship between ROS and glucose uptake, but on what grounds this can be determined, citing the results of human clinical trials as well as HepG2 cells.

Response 1: Thanks for the valuable suggestion. The relationship between ROS and glucose uptake should be discussed and revised as suggested. In this study, we applied the method of high glucose (30 mM) and high fat (0.2 mM palmitate) treatment to establish insulin resistance cell model (Chen et al., 2017). It’s well known that hepatic insulin resistance was characterized by depressed glucose consumption and storage (Wilson et al., 2003; Joseph et al., 2022), and the intracellular ROS levels rose significantly under insulin resistance status (Tan et al., 2019; Yan et al., 2017). Previous studies in diabetic patients found that hyperglycemia reduced the entry of glucose into the pentose phosphate pathway (Zhang et al., 2000), thereby reducing NADPH content and activating the NADPH oxidases that generate ROS (Drummond et al., 2014; Eelen et al., 2015). In addition, the endothelial nitric oxide synthase (eNOS) would generate ROS instead of NO when the availability of NO precursors or cofactors were reduced, including NADPH (Kawashima et al., 2004; Eelen et al., 2015). Similar results were observed in HepG2 cells, the ROS levels were elevated in high glucose and fat environments (Cang X et al, 2016; Ma SB et al., 2017). Therefore, close relationship existed between glucose metabolism and ROS generation, while the pentose phosphate pathway (PPP) impairment played important roles in the process. The possible relationship between ROS and glucose uptake was discussed in the revised manuscript.

  1. I feel that the data are not sufficient to determine that increased expression of the HKDC1 and G6PC1 genes is associated with increased glucose uptake in HepG2 cells. Not only phenomenology, but also a variety of examples and the increase in the number of proteins involved should be investigated, and it is necessary to fully reconsider whether such a conclusion can be drawn from this experiment alone.

Response 2: We appreciated reviewer’s suggestion. Yes, we fully agree that the data in current study are not sufficient to determine that increased expression of the HKDC1 and G6PC1 genes associated with increased glucose uptake in HepG2 cells. It’s well known that glucose uptake is closely related to its metabolism, while glucose metabolism is incredible complicated and connected to numerous other biological reactions in the cell. Therefore, we applied the genome wide RNAseq method in this study, trying to clarify the possible regulatory mechanism and target genes related to the VLPVPQK improved glucose uptake. Following the normal protocol commonly used for sequencing data analysis, we found 69 differentially expressed genes between model and VLPVPQK groups. After functional enrichment analysis, 7 genes (HKDC1, TRIB3, GNE, AKR1B10, CA9, G6PC1, and NDUFA4L2) might participate in the regulation of glucose metabolism. Theoretically, all the above 7 genes should be analyzed. In this study, we focused on three genes, including HKDC1, G6PC1, and AKR1B10. Because HKDC1 and G6PC1 were core genes that had the largest number of connections with carbohydrate metabolism related pathways, and AKR1B10 showed showed relative high abundance in HepG2 cells. Although lots of studies reported that the HKDC1 and G6PC1 exerted important roles in regulating glucose utilization (Ludvik AE et al., 2016; Khan MW et al., 2018; Singh P et al., 2017; Guo C et al., 2015; Karthi S et al., 2017), the function of other proteins should be considered. In addition, the difference existed among the gene expression, protein expression, and enzyme activity should be taken into consideration. Overall, there are some limitations in our study as mentioned above, and the conclusion should be modified. The limitation of the current study was added and the conclusion was reorganized in the revised manuscript.

References:

Chen, Y.Q., Yan, X., Chen, M.J., Lin, L., Yang, C.F., Qiu-Zhe, L.I., Liu, B., Zhao, C. Anti-diabetic activity of alcohol extracts from lessonia nigrescens and its effects on intestinal microflora in mice. Biotechnol. Bull. 2017, 33, 162–169.

Wilson JE. Isozymes of mammalian hexokinase: structure, subcellular localization and metabolic function. J Exp Biol. 2003, 206(Pt 12):2049-57.

Joseph L. Zapater,Kristen R. Lednovich, Md. Wasim Khan, Carolina M. Pusec, and Brian T. Layden. Hexokinase domain-containing protein-1 in metabolic diseases and beyond. Trends in Endocrinology & Metabolism, 2022, 33, No. 1.

Tan, Y; Jin, Y; Wu, X; et al. PSMD1 and PSMD2 regulate HepG2 cell proliferation and apoptosis via modulating cellular lipid droplet metabolism.BMC Molecular Biology, 2019, 20(1), 24.

Yan, F., Zheng, X., Anthocyanin-rich mulberry fruit improve insulin resistance and protects hepatocytes against oxidative stress during hyperglycemia by regulating AMPK/ACC/mTOR pathway. Journal of Funtional Foods 2017, 30, 270-281.

Zhang Z, Apse K, Pang J, Stanton RC. High glucose inhibits glucose-6-phosphate dehydrogenase via cAMP in aortic endothelial cells. J Biol Chem. 2000, 275(51):40042-7.

Drummond GR, Sobey CG. Endothelial NADPH oxidases: which NOX to target in vascular disease? Trends Endocrinol Metab. 2014, 25(9):452-63.

Eelen G, de Zeeuw P, Simons M, Carmeliet P. Endothelial cell metabolism in normal and diseased vasculature. Circ Res. 2015, 116(7):1231-44.

Kawashima S. Malfunction of vascular control in lifestyle-related diseases: endothelial nitric oxide (NO) synthase/NO system in atherosclerosis. J Pharmacol Sci. 2004, 96(4):411-9.

Cang X, Wang X, Liu P, Wu X, Yan J, Chen J, Wu G, Jin Y, Xu F, Su J, Wan C, Wang X. PINK1 alleviates palmitate induced insulin resistance in HepG2 cells by suppressing ROS mediated MAPK pathways. Biochem Biophys Res Commun. 2016, 478(1):431-438.

Ma SB, Zhang R, Miao S, Gao B, Lu Y, Hui S, Li L, Shi XP, Wen AD. Epigallocatechin-3-gallate ameliorates insulin resistance in hepatocytes. Mol Med Rep. 2017, 15(6):3803-3809.

Ludvik AE, Pusec CM, Priyadarshini M, Angueira AR, Guo C, Lo A, Hershenhouse KS, Yang GY, Ding X, Reddy TE, Lowe WL Jr, Layden BT. HKDC1 Is a Novel Hexokinase Involved in Whole-Body Glucose Use. Endocrinology. 2016, 157(9):3452-61.

Khan MW, Ding X, Cotler SJ, Clarke M, Layden BT. Studies on the Tissue Localization of HKDC1, a Putative Novel Fifth Hexokinase, in Humans. J Histochem Cytochem. 2018, 66(5):385-392.

Singh P, Han EH, Endrizzi JA, O'Brien RM, Chi YI. Crystal structures reveal a new and novel FoxO1 binding site within the human glucose-6-phosphatase catalytic subunit 1 gene promoter. J Struct Biol. 2017, 198(1):54-64.

Guo C, Ludvik AE, Arlotto ME, Hayes MG, Armstrong LL, Scholtens DM, Brown CD, Newgard CB, Becker TC, Layden BT, Lowe WL, Reddy TE. Coordinated regulatory variation associated with gestational hyperglycaemia regulates expression of the novel hexokinase HKDC1. Nat Commun. 2015, 6:6069.

Karthi S, Rajeshwari M, Francis A, Saravanan M, Varalakshmi P, Houlden H, Thangaraj K, Ashokkumar B. 3'-UTR SNP rs2229611 in G6PC1 affects mRNA stability, expression and Glycogen Storage Disease type-Ia risk. Clin Chim Acta. 2017, 471:46-54.

Best regards,

Xinyan Peng

Reviewer 2 Report

In the manuscript ‘The effects and regulatory mechanism of casein derived peptide VLPVPQK in alleviating insulin resistance of HepG2 cells’ authors explore the effects and regulatory mechanism of the peptide VLPVPQK on metabolic disorder of liver. Although the manuscript is well written, neat, and the material and methods well explained, in the reviewer's opinion, the manuscript has criticism.

Comments:

The English throughout the article must be widely checked. Countless grammatical and typographical errors (commas and spaces)

the (%) on the y-axis of Figure 1A should be separated from the rest

The figure captions should specify the standard error of the mean, the n, and the statistical test used.

The material and methods are confusing and should be vastly improved.

When carrying out the antioxidant activity, the cells were treated with the peptide at different concentrations. During how much time? was the supernatant removed? were the cells lysed to measure antioxidant activity? If they were not lysed and the supernatant was collected, the antioxidant activity of only the peptide is being measured.

The order of materials and methods and results is confusing. It appears that the antioxidant activity was done in vitro, but then these graphs are attached to the viability graph. I advise starting the material and methods with cells and continuing with viability. In the same way it should be ordered in the results.

Author Response

Thanks so much for your hard work on our manuscript "foods-2435865". We thank the reviewers for thoroughly reviewing our manuscript and making thoughtful comments. The manuscript has been revised carefully, and the detailed corrections requested by the reviewers have been done and listed below point by point.

  1. The English throughout the article must be widely checked. Countless grammatical and typographical errors (commas and spaces)

Response 1: Really appreciate your valuable comments and sorry for the errors. We carefully went though the manuscript and revised as suggested, and the font colour of revised parts were marked in red in the text.

  1. the (%) on the y-axis of Figure 1A should be separated from the rest

Response 2: Thanks for your careful check. We are sorry for our carelessness. Revised as suggested. 

  1. The figure captions should specify the standard error of the mean, the n, and the statistical test used.

Response 3: Thanks so much for your valuable suggestion, and revised as suggested.

  1. The material and methods are confusing and should be vastly improved.

When carrying out the antioxidant activity, the cells were treated with the peptide at different concentrations. During how much time? was the supernatant removed? were the cells lysed to measure antioxidant activity? If they were not lysed and the supernatant was collected, the antioxidant activity of only the peptide is being measured.

Response 4: Thanks so much for your valuable comments, and revised the materials and methods carefully.

When carrying out the antioxidant activity, the cells were treated with the peptide at different concentrations for 48 hour. Then the treatment culture supernatant was removed. The antioxidant activity was measured using synthesized peptide instead of cells.

  1. The order of materials and methods and results is confusing. It appears that the antioxidant activity was done in vitro, but then these graphs are attached to the viability graph. I advise starting the material and methods with cells and continuing with viability. In the same way it should be ordered in the results.

Response 5: Thanks so much for your helpful suggestions. Yes, the materials, method, and results is confusing in this format. We reorganized them in the revised manuscript and the font colour of revised parts were marked in red in the text. In this study, we firstly detected the antioxidant activity of the synthesized peptide VLPVPQK by detecting the oxygen radical absorbance and hydroxyl radical scavenging capacity. Then, the in vitro cell culture experiments were conducted. The major revision made was as follow:

Line 87 “2.2. In vitro Assay of the antioxidant activity of VLPVPQK” was changed to “2.2. Assay of the antioxidant activity of VLPVPQ”.

Figure 1C was moved to Figure 2A. Then, Figure 1 represented the antioxidant results, while Figure 2 showed the cell culture data.

Best regards,

Xinyan Peng
